# Biodiversity of Secondary Metabolites Compounds Isolated from Phylum Actinobacteria and Its Therapeutic Applications

**DOI:** 10.3390/molecules26154504

**Published:** 2021-07-26

**Authors:** Muhanna Mohammed Al-shaibani, Radin Maya Saphira Radin Mohamed, Nik Marzuki Sidik, Hesham Ali El Enshasy, Adel Al-Gheethi, Efaq Noman, Nabil Ali Al-Mekhlafi, Noraziah Mohamad Zin

**Affiliations:** 1Micro-Pollutant Research Centre (MPRC), Faculty of Civil Engineering and Built Environment, Universiti Tun Hussein Onn Malaysia, Parit Raja 86400, Johor, Malaysia; muhanna@uthm.edu.my; 2Center for Diagnostic, Therapeutic and Investigative Studies, Faculty of Health Sciences, Universiti Kebangsaan Malaysia, Kuala Lumpur 50300, Malaysia; noraziah.zin@ukm.edu.my; 3Faculty of Agro-Based Industry, Universiti Malaysia Kelantan, Jeli 17600, Kelantan, Malaysia; 4Institute of Bioproducts Development (IBD), Universiti Teknologi Malaysia (UTM), Skudai 81310, Johor, Malaysia; henshasy@ibd.utm.my; 5City of Scientific Research and Technology Applications (SRTA), 21934 New Burg Al Arab, Alexandria, Egypt; 6Applied Microbiology Department, Faculty of Applied Sciences, Taiz University, Taiz 6803, Yemen; hw160052@siswa.uthm.edu.my; 7Atta-ur-Rahman Institute for Natural Product Discovery, UiTM, Puncak Alam Campus, Bandar Puncak Alam 42300, Selangor, Malaysia; nabilali7@tu.edu.ye; 8Biochemical Technology Program, Department of Chemistry Faculty of Applied Science, Thamar University, Thamar P.O. Box 87246, Yemen

**Keywords:** microbial ecology, aquatic and marine environments, drug-resistant pathogens, *Streptomyces*, natural products, VOSviewer software

## Abstract

The current review aims to summarise the biodiversity and biosynthesis of novel secondary metabolites compounds, of the phylum Actinobacteria and the diverse range of secondary metabolites produced that vary depending on its ecological environments they inhabit. Actinobacteria creates a wide range of bioactive substances that can be of great value to public health and the pharmaceutical industry. The literature analysis process for this review was conducted using the VOSviewer software tool to visualise the bibliometric networks of the most relevant databases from the Scopus database in the period between 2010 and 22 March 2021. Screening and exploring the available literature relating to the extreme environments and ecosystems that Actinobacteria inhabit aims to identify new strains of this major microorganism class, producing unique novel bioactive compounds. The knowledge gained from these studies is intended to encourage scientists in the natural product discovery field to identify and characterise novel strains containing various bioactive gene clusters with potential clinical applications. It is evident that Actinobacteria adapted to survive in extreme environments represent an important source of a wide range of bioactive compounds. Actinobacteria have a large number of secondary metabolite biosynthetic gene clusters. They can synthesise thousands of subordinate metabolites with different biological actions such as anti-bacterial, anti-parasitic, anti-fungal, anti-virus, anti-cancer and growth-promoting compounds. These are highly significant economically due to their potential applications in the food, nutrition and health industries and thus support our communities’ well-being.

## 1. Introduction

Global demand for new chemotherapeutic compounds and antibiotics with high bioactivity and low toxicity has increased recently due to the emergence of life-threatening microorganisms and multidrug resistance agents among viruses, bacteria and fungi [1]. Additionally, the detection of secondary metabolites molecules with unique modes of action established various therapeutic agents’ strategies for treating many illnesses [2]. To be more specific, endophytic Actinobacteria are microorganisms that represent a new production source of a large number of secondary metabolites, including alkaloids, beta-lactams, sulfonamides, aminoglycosides, glycopeptides, siderophores, quorum-sensing molecules, immunosuppressants, polyene macrolides, saccharides, pyrazoloisoquinolinones, butenolides, nucleosides and degradative enzymes [3]. In fact, it has been reported that more than 10,000 various bioactive compounds have been discovered from Actinobacteria [4].

Endophytic microbes refer to a group of microorganisms, mostly fungi and bacteria, that exist in the host plant’s intracellular space. It usually causes no obvious harmful effect or symptoms of the disease and could produce various associations such as trophobiotic communalistic, mutualistic and symbiotic co-existence [5]. Endophytes in woody plant hosts could exist within host tissues and protect host plants against herbivores and other pathogenic microorganisms [6]. ActinobacteriaActinobacteria are Gram-positive bacteria with high guanine and cytosine (G + C) content in their genomes, and they are classified into 6 classes, 79 families of 46 orders and 10 fresh families of 16 new orders based on phylogeny using 16S rRNA sequences. The Actinobacterial classes consist of *Thermoleophilia, Rubrobacteria Nitriliruptoria, Coriobacteria, Actinomycetia* and *Acidomicrobiia* Salam et. al. [7]. Actinobacteria have ubiquitous characteristics. They are present in diverse ecosystems on the earth such as endophytically with plants and in terrestrial and aquatic environments. An abundance of Actinobacteria species have been recorded in ordinary, extraordinary and extreme environments with high or low temperatures, high radiation, acidic/alkaline pH, salinity, low levels of available moisture and nutrients [8].

The genus *Streptomyces* is a Gram-positive bacteria. It is the largest genus of the phylum Actinobacteria, which has complex growth and can produce various secondary metabolites [8]. In addition, there are more than 800 *Streptomyces* species that have been found to date (see http://www.bacterio.net/*Streptomyces*.html (accessed on 20 August 2020) [9]. *Streptomyces* is the major microbial genus of the most antibiotic-producing bacteria in the microbial world discovered so far, where streptomycin, gentamycin, rifamycin, chloramphenicol and erythromycin are produced by *Streptomyces* [10].

Actinobacteria have a large number of secondary metabolite biosynthetic gene clusters. Biosynthetic gene clusters (BGCs) are known as genes comprising locally clustered groups encoding a secondary metabolite biosynthetic pathway. In addition, BGCs contain genes encoding all enzymes required to produce secondary metabolites and pathway-specific regulatory genes. The Actinobacteria have diverse physiology and metabolic flexibility with high potential to produce novel bioactive compounds and enzyme production [11].

The current review aims to summarise the biodiversity, biosynthesis of novel bioactive secondary metabolites, detection of new resources, and strategies to search for potential bioactive compounds producers. In addition, it aims to determine the effects of environments and ecosystem on the phylum Actinobacteria that produce potential bioactive compounds in the pharmaceutical industry market. The VOSviewer software tool was used to visualise bibliometric networks to support Scopus’s prevalent bibliographic databases. This information would be helpful to other scholars who attempt to discover and isolate specific bioactive compounds from phylum Actinobacteria under different ecosystems. Besides their utilisation by people, the perception of the function and distribution of microbial products is regarded as significant to understand microbial families and their impact on biogeochemical cycles.

## 2. Methods and Protocol

### 2.1. Study Design and Search Strategy

The methodology of this review was conducted as demonstrated in Figure 1. The process involved six main steps: Step 1 was an identification of the target articles through the Scopus database in the period between 2010 and 22 March 2021 using the general keywords “Actinomycetes” OR “Actinobacteria,” OR “*Streptomyces*”. The number of articles obtained was *n* = 70,618. In Step 2, the resulting articles were further screened using more specific keywords, including “Natural Products” OR “Secondary Metabolites” (*n* = 15,141). Then, in Step 3, from that number of articles, further screening was carried out using more specific keywords, which included antibiotics, anti-fungal, anti-cancer, anti-parasitic, “antimalarial,” OR anti-inflammatory (*n* = 13,041). In Step 4, from the preceding number of articles, screening was implemented using more precise keywords, such as endophytic, thermophilic, halophilic OR marine (*n* = 5435 was extracted). Subsequently, in Step 5, from the total number of articles, further screening was considered using more specific keywords, including “mechanism of action” OR “mode of action” (*n* = 985). Finally, in Step 6, from these articles, the final screening was executed using more specific keywords “drug-resistance” OR “resistant”. The number of articles based on keywords is (*n* = 436), extracted studies based on related titles was *n* = 336; reviewed studies out of limitations were *n* = 31; records excluded based on the validity of the study data and clear contributions were *n* = 260, and out of limitations were *n* = 46. Finally, the number of studies included in this review was *n* = 116. A review scheme was conducted to determine all research documents published only in the English language, presented in Figure 1.

### 2.2. Data Analysis

The VOSviewer software was used to visualise the bibliometric networks to build assistance for common bibliographic databases from the Scopus database. The most important journals, articles, authors, organisations and states among such articles were identified. Besides this, bibliographic coupling and most utilised keywords in the abstracts with keywords and titles were identified.

The majority of diversity keywords from the reviewed and cited papers based on the Scopus database were *Streptomyces*, Actinobacteria, natural products, primary, secondary metabolites, habitat effects of environments, pharmaceutical industry, as shown in Figure 2A.

This analysis includes diverse secondary metabolite molecules induced by environmental factors such as antibiotic agents, biological products, antineoplastic agent, anti-fungal, and agent with antimalarial activity, as shown in Figure 2B. The reviewed and cited papers based on the scattered keywords of the antimicrobial isolated from 2010 to 2021 from phylum Actinobacteria were vancomycin, polyketide, tetracycline, cyclopeptide, erythromycin, streptomycin, macrolides and amphotericin B, as represented in Appendix A. The percentage of keywords diversification from the Scopus database for *Streptomyces*, Actinobacteria, natural products, primary, secondary metabolites, habitat effects of environments, pharmaceutical industry and anti-infective agents occurrence using VOSviewer software tool to analyse and visualise scientific literature is shown in Appendix A. The most reviewed and cited papers based on the scattered keywords of the countries for the publication and citation were USA (46 %), UK and India (13% each), Germany (11%), China (6%) and Thailand (3%), while KSA, Egypt, Pakistan, Malaysia had only 2% each, as shown in Figure 2C and Appendix A.

## 3. Primary and Secondary Metabolites Natural Products

Actinobacteria synthesise diverse metabolite molecules that have key roles in their heterogeneous and complex microenvironments. Natural products, also known as sec-ondary metabolites, are useful compounds developed by microbes. These are not usually needed for natural cell development, but they provide advantages to the cells in other ways [12]. Such compounds could play roles in inhibition, communication, nutrient ac-quisition, or other associations with nearby environments or organisms.

Natural products have been utilised for a long time. In fact, the Chinese are amongst users of traditional medicines with more than five thousand plants and microbes’ products in their pharmacopoeia [13]. Therapeutic plant species have been, and are still being, used in traditional medicine in several countries [12]. Primary metabolites are chemicals required for ordinary growth, development and reproduction of organisms as well as maintenance of cellular function, representing the key role in the survival of organisms. Besides this, the primary metabolite plays an active role in the anabolic and catabolic processes in many organisms or cells [14]. The secondary metabolites are substances formed during the end or near the stationary stage of organisms’ development. They are very significant for nutrition and health and are, therefore, economically important [14]. Even though they serve different survival actions in nature, they do not necessarily play a critical role in the growth and development of the organism producing them [15].

## 4. History of Isolation of Secondary Metabolites from Actinobacteria

Historically, in (1940), Waksman and Woodruff isolated actinomycin D from soil bacteria [16]. Then, Schatz et al. in 1944 [17] isolated streptomycin, an effective antibiotic against tuberculosis. Furthermore, hundreds of various antibiotics were reported from the genus *Streptomyces* [18]. Actinobacteria are among the secondary metabolites producers and hold high pharmacological and commercial interest. It has great capability to produce secondary metabolites such as immunomodulators [19], antibiotics [20], anti-cancer drugs [21], growth factors [22], anthelminthic enzymes and herbicides [23]. Appendix A describes the historical isolation of bioactive compounds from Actinobacteria from the first isolation by Selman Waksman [20].

Several anti-fungal, anti-parasitic, bioactive compounds, growth-promoting compounds and anti-cancer compounds with their chemical classification and application isolated from Actinobacteria and *Streptomyces* sp. are represented in Appendix A. The molecular structures of several bioactive compounds isolated from Actinobacteria and *Streptomyces* are demonstrated in Figure 3, Figure 4 and Figure 5.

## 5. Microbial Ecology of Actinobacteria

The diversity of Actinobacteria has been investigated in several special or extreme environments, such as Actinobacteria in terrestrial environments, Actinobacteria in aquatic and marine environments, and thermophilic as well as alkaliphilic and haloalkaliphilic Actinobacteria.

### 5.1. Actinobacteria in Terrestrial Environments

Terrestrial Actinobacteria have several antimicrobial capabilities, including three active compounds determined as 2,3-heptanedione, butyl propyl ester and cyclohexane with an antimalarial activity isolated from *Streptomyces* SUK 08 [24].

The diketopiperazines compounds and chloramphenicol were isolated from *Streptomyces* SUK 25 isolated from the *Zingiber spectabile* plant in Malaysia [25]. Meanwhile, *Streptomyces* sp. CAH29 isolated from the rhizosphere environment can produce tetrangomycin, possessing potent anti-bacterial and anti-fungal action against *Candida albicans*, methicillin-resistant *Staphylococcus aureus*, *Streptococcus pyogenes* and *Staphylococcus aureus* having 14, 10, 12 and 8 mm inhibition zones, respectively [26]. Besides this, the isolate revealed a marked anti-tumour action with 1.1 μg/mL and IC_50_ 3.3 against colorectal (HCT116) and hepatocellular (HepG2) carcinoma cell lines, respectively [27]. Actinomycin D, which has anti-tumour and anti-bacterial activity, was extracted from *Streptosporangium* sp. (AI-21) at Hardwar district in Uttarakhand state, India [28]. The list of the terrestrial and rhizosphere Actinobacteria isolation and screening for their antimicrobial activity are shown in Appendix A.

### 5.2. Actinobacteria in Aquatic and Marine Environments

Marine Actinobacteria are abundant sources for marine drug discovery causal for many bioactive compounds of biomedical appearance. Almost 30 strains of actinomycetes were separated and determined from various families of genus *Streptomyces*. The *Streptomyces* isolates of M93, W108, W38, M72, M71 and M1 are amongst all the chosen strains revealed to show potent antimicrobial action against the multidrug-resistant bacteria (MDRB) [29]. Furthermore, antitumor compounds produced by marine Actinobacteria is reported [30]. Moreover, antimalarial activity was also described from marine Actinobacteria as bioactive compounds [28]. Additionally, from freshwater sediments, 84 Actinobacteria were separated and isolated into a prevalent genus *Streptomyces* as well as eight uncommon genera such as *Micrococcus, Kocuria, Nocardiopsis, Promicromonospora, Saccharopolyspora, Amycolatopsis, Prauserella,* and *Rhodococcus*. All strains revealed significant inhibition potentials against yeast pathogens, Gram-negative bacteria and Gram-positive bacteria [5].

Moreover, *Saccharopolyspora* sp. IMA1 was separated from the coral reef environment. Its metabolites existed in aquatic bacterial pathogens *Vibrio vulnificus, Vibrio parahemolyticus* and *Vibrio harveyi*. The IMA1 crude extract revealed excellent antioxidant activity, where 2,2′-azino-bis 3-ethylbenzothiazoline-6-sulfonic acid (ABTS) has a radical scavenging activity [31]. Almost 52 strains of Actinobacteria were isolated from the marine sediment of the Portuguese coast. The isolates’ greatest fraction were from the genus *Micromonospora*, from which six *Streptomyces* strains were active against *Candida. albicans*, with minimum inhibitory concentration (MIC) values ranging from 3.90 to 125 µg mL^−1^ [2].

The list of bioactive compounds isolated from aquatic and marine Actinobacteria are demonstrated in Table 1.

### 5.3. Thermophilic Actinobacteria

Thermophilic Actinobacteria can withstand temperatures in the range of 40–80 °C [8]. They thrive on organic debris that has decomposed (dead plant and animal materials). Thermophilic Actinobacteria are divided into two types: moderately thermophilic and strictly thermophilic. Certain strains of moderately thermophilic Actinobacteria may develop at temperatures ranging from 37 to 65 °C, whereas some grow between 28 and 60 °C and require 45–55 °C for maximal growth. However, maximum proliferation happens at 55–60 °C.

Thermotolerant Actinobacteria, on the other hand, can withstand temperatures as high as 50 °C [38]. Determining fresh bioactive compounds from taxonomically sole strains of extremotrophic or extremophilic Actinobacteria resulted in the anticipation that mining such groups might provide an alternate dimension to the route of subordinate metabolite resources [39]. Thermotolerant and thermophilic Actinobacteria are unique for having specific metabolic rates and physical features, which are useful in several ecological functions. It was also reported that thermotolerant Actinobacteria like *Streptosporangium sp., S. lanatus, S. coeruleorubidis, S. toxytricini* and *Streptomyces tauricus,* are claimed to inhibit the rhizosphere of several plants in the Kuwait desert during hot seasons [11]. Thermophilic Actinobacteria have not been widely investigated. However, they have developed significant antibiotics like thermomycin from *Streptomyces thermophilus* and anthramycin, an anti-tumour drug created by *Streptomyces refuineus* [40].

The *Streptomyces* strain Al-Dhabi-2 found in Saudi Arabia’s thermophilic region indicated antimicrobial capabilities against pathogenic microorganism [41]. Three strains identified as DJT 15 *Streptomyces thermoviolaceus* subsp. apingens, DJT 32 *Saccharomonospora viridis*, and DJT *36 Saccharomonospora glauca* exposed inhibitory activity in bioassays with respect to the *Enterobacter* species (ESKAPE) pathogens, *Pseudomonas aeruginosa*, *Acinetobacter baumannii Klebsiella pneumoniae, Staphylococcus aureus* and *Enterococcus faecium.* Apart from that, DJT 32 and 36 prevented the growth of *Aspergillus fumigatus,* a filamentous fungus isolated from the same compost. Azole-resistant human pathogenic strains of *Aspergillus fumigatus* were also shown to be inhibited by strain DJT 32 [42].

### 5.4. Alkaliphilic and Haloalkaliphilic Actinobacteria

An alkaliphilic and mildly halophilic bacterial strain develops optimally at pH ranging from 9.0 to 10.0, with 5–7% (*w*/*v*) NaCl [43]. Haloalkaliphilic ActinobacteriaActinobacteria are found in pristine coastal habitat in the saline soil. These unique ActinobacteriaActinobacteria are hereditarily different, developing at high salt concentrations [44]. Alkaliphilic ActinobacteriaActinobacteria are, thus, classified into three main categories: alkali-tolerant ActinobacteriaActinobacteria (developed in the pH range between 6 and 11, abstemiously alkaliphilic (developed in a pH ranging from 7 to 10 but display weak development at pH 7.0), and alkaliphilic (grow optimally at pH 10–11) [11].

Moreover, *Streptomyces clavuligerus* (strain Mit-1) was isolated from Mithapur (Western Coast, Gujarat, India) and is described as a salt-tolerant alkaliphilic actinomycete. Note that this strain secreted alkaline protease [45]. Besides this, Thakrar et al. [46] issued the capability to produce protease from the two halo-tolerant and alkaliphilic actinomycetes extracted from the salt-enriched soil of Coastal Gujarat in India. These strains are identified as *Nocardiopsis alba* OM-4 and *Nocardiopsis alba* TATA-13. In addition, the phylogeny and multiplicity of the haloalkaliphilic ActinobacteriaActinobacteria study represent a concise and comprehensive account utilising conventional and molecular techniques. This includes the occurrence and cultivation of the marine actinomycetes from the seawater and sediments of pristine coastal habitat in Gujarat (India).

They found that the ActinobacteriaActinobacteria belonged to *Prauseria*, *Streptomyces*, *Brachybacterium* and *Nocardiopsis* [47]. Besides this, the multiplicity of alkali tolerant ActinobacteriaActinobacteria in numerous soils is gathered from various Algerian Sahara areas. Almost 29 alkali-tolerant ActinobacteriaActinobacteria strains were separated by utilising a complex agar medium. The strains were then tested for non-ribosomal peptide synthetases and genes encoding polyketide synthases, as well as antibiotic behaviour against a broad microorganisms’ variety. They found that some strains can create subordinate metabolites against several pathogenic microbes [19].

A list of bioactive compounds isolated from thermophilic, alkaliphilic and haloalkaliphilic ActinobacteriaActinobacteria is shown in Table 2.

## 6. The Important Antibiotics Isolated from Actinobacteria against Drug-Resistant Pathogens

Yücel and Yamaç [50] showed that extracts from *Streptomyces* sp. 1492 exhibited antimicrobial activity against MRSA, Vancomycin-resistant *Enterobacter faecium* (VRE) and *Acinetobacter baumanii* with 125 µg/mL and 250 to 1000 µg/mL of MICs and MBCs, respectively. From *Verrucosispora* sp. strains isolated from the Japan Sea’s sediment, Goodfellow and Fiedler [51] identified atrop-abyssomicin C and proximicins A, B and C. It inhibits the formation of p-aminobenzoate during tetrahydrofolate synthesis in Gram-positive bacteria with methicillin-resistant *Staphylococcus aureus* (MRSA). The *Streptomyces malachitofuscus* has anti-fungal action against *Candida albicans* and *Mucor miehei*, as Sajid et al. [52] reported. The spoxazomicins was extracted from *Streptosporangium oxazolinicum* sp. nov., obtained from the root of numerous orchids at Okinawa. The study also isolated spoxazomicins A-C from *Streptosporangium oxazolinicum* K07-0460 (T) with antitrypanosomal activity [53]. Besides this, the arylomycine isolated from *Streptomyces* sp. HCCB10043 show activity against Gram-positive bacteria, among which included *Staphylococcus aureus* [54]. The chlorocatechelins A and B extracted from *Streptomyces* sp. are fresh siderophores encompassing an acylguanidine structure and chlorinated catecholate sets that inhibit various bacteria and fungi [55]. An extremely potent subordinate metabolite formed by endophytic strain is known as *Streptomyces* sp. HUST012. It showed antimicrobial and anti-tumour activities separated from the medicinal plant stems, *Dracaena cochinchinensis Lour* [56]. The updated list of antibiotics separated from ActinobacteriaActinobacteria and *Streptomyces* is given in Table 3.

The existing literature reported 124 compounds from Actinobacteria with anti-MRSA activity. Several bioactive pharmaceutical compounds generated by Actinobacteria was found to be effective against Vancomycin-resistant *Enterococcus* (VRE), Methicillin-resistant *Staphylococcus aureus* (MRSA) and other drug-resistant bacterial pathogens strains. Kakadumycin A was isolated from *Streptomyces* sp. NRRL 30566 inhibited the growth of MRSA American Type Culture Collection (ATCC) 33591 four times lower in MIC than vancomycin at a concentration of 0.5 µg/mL compared to the concentration of vancomycin of 2.0 μg/mL, as reported by Castillo et. al. [57]. In addition, polyketide antibiotic SBR-22 comes from a fresh actinomycetes strain, labelled as BT-408, a novel strain of *Streptomyces psammoticus.* The strain showed anti-bacterial activity against MRSA, as documented by Sujatha et. al. [58]. Besides this, Yoo et. al. [59] isolated laidlomycin from *Streptomyces* CS684 obtained from the soil at Jeonnam, Korea, with potential activity against VRE. Meanwhile, Laidi et. al. [60] isolated a new actinomycetes strain designated as SK4-6 from Egyptian soil that demonstrated strong activity against bacteria, including MRSA and *Micrococcus luteus.* Furthermore, *Streptomyces* sp. PVRK-1 was isolated from the Manakkudy mangroves in Tamil Nadu, South India. This *Streptomyces* has been able to produce many small molecules exhibiting potent anti-MRSA activity [43].

**Table 3 molecules-26-04504-t003:** The list of important producers and chemical classification of antibiotics isolated from Actinobacteria.

Antibiotic	Producer	Chemical Class	Reference
Cephamycin C	*Nocardia lactamdurans*	B- Lactam	[61]
Chlortetracycline	*S. aureofaciens*	Tetracycline	[62]
Clavulanic acid	*S. clavuligerus*	B- Lactam	[63]
Cycloserine	*S. orchidaceus*	Peptide	[64]
Daptomycin	*S. rodeosporus*	Lipopeptide	[65]
Daunorubicin	*S. Peucetius*	Peptide	[66]
FK506	*S.tubercidicus*	Macrolide	[67]
Fortimicin	*Micromonospora olivasterospora*	Aminoglycoside	[68]
Fosfomycin	*S. fradiae*	Phosphoric acid	[69]
Fumaramidmycin	*S. kurssanovii*	Alkaloids	[70]
Gentamycin	*Micromonospora spp*	Aminoglycoside	[71]
Kanamycin	*S. kanamyceticus*	Aminoglycoside	[72]
Lincomycinn	*S. lincolnensis*	Sugar—amide	[73]
Neomycin	*S. fradiae*	Aminoglycoside	[74]
Nikkomycin	*S. tendae*	Nucleoside	[75]
Nocardicin	*Nocardia uniformis*	B- Lactam	[76]
Novobiocin	*S. neveus*	Aminocoumarin	[77]
Oleandomycin	*S. antibioticus*	Macrolide	[78]
Oxytetracycline	*S. rimosus*	Tetracycline	[79]
Paromomycin	*S.rimosus forma*	Aminoglycoside	[80]
Rifamycin	*Amycolatopsis Ansamycin*	RNA polymerase (PK)	[81]
Spiramycin	*S. ambofaciens*	Macrolide (PK)	[82]
Streptomycin	*S. griseus*	Aminoglycoside	[83]
Tetracycline	*S. aureofaciens*	Tetracycline (PK)	[84]
Thienamycin	*S. cattleya*	β-Lactam Peptidoglycan	[85]
Tobramycin	*S. tenebrarius*	Aminoglycoside	[86]
Vancomycin	*S.orientalis*	Peptidoglycan	[87]

Abbreviation: *S.* = (*Streptomyces).*

In addition, Choi et. al. [88] claimed that *Streptomyces* sp. CS392 have anti-bacterial activity against VRE and MRSA. Additionally, the strain *Streptomyces* sp (VITBRK2) can yield strong antimicrobial activity against VRE and MRSA strains [89]. Moreover, *Streptomyces rubrolavendulae* ICN3 strain displayed potent antimicrobial activity towards MRSA, with a 42 mm inhibition zone. The MIC was 500 μg/mL from the crude extracts, while the cleansed compound was identified as C23 with MIC 2.5 μg/mL in the in vitro assay [90]. Similarly, from a mangrove forest soil on Peninsular Malaysia’s east coast, the MUSC 135T strain was isolated. It can produce broad-spectrum antibiotic bacitracin against MRSA strain ATCC BAA-44 [91]. Correspondingly, from the stems of *Dracaena Cochinchinensis Lour* plant, *Streptomyces* sp. HUST012 was isolated, producing two potent antimicrobial compounds towards MRSA ATCC 25923 with MIC of 62.5 and 0.04 µg/mL, identified as SPE-B11.8-B5.4 [56]. *Streptomyces* sp. KB1 from Ao-Nang, Krabi province, Thailand, contains a compound known as 2,4-Di-tert-butylphenol, an anti-MRSA MIC of 31.15 µg/mL in comparison to vancomycin, with a MIC value of 1.56 µg/mL [92]. Diketopiperazines and chloramphenicol extracted from *Streptomyces* SUK 25, derived from the *Zingiber spectabile* root, seem effective towards MRSA [25]. Moreover, Bhakyashree and Kannabiran [93] reported the separation of anti-MRSA compounds from *Streptomyces* sp. VITBKA3 strain proteins and cell membrane biosynthesis inhibitors. The strains of *Streptomyces* sp. derived from different environments with potentials in production of anti-MRSA bioactive compounds from 2014 until 2020 is given in Table 4.

## 7. Production of Enzymes from Actinobacteria

Both marine and terrestrial Actinobacteria in Table 5 produce a wide range of biologically active enzymes. They can secrete protease, cellulases, lipase, xylanase, pectinase and amylase. These enzymes are vitally important in textile or paper industries, the food industry and fermentation, besides biotechnological application. Consequently, Actinobacteria have been discovered to be a good L-asparaginase source. Various Actinobacteria often those extracted from soils, for example, *Nocardia* sp *Streptomyces albidoflavus*, *S. griseus* and *Streptomyces karnatakensis* produce several enzymes [102]. Enzymes like urease, chitinase and catalase are also generated from Actinobacteria [103]. Surprisingly, keratinase, an enzyme that destroys the feathers of poultry chickens, has been successfully produced from *Nocradiopsis* sp. [104]. Likewise, Actinobacteria separated from goat and chicken gut revealed the existence of numerous enzymes like lipase, phytase, protease and amylase [105].

## 8. Mechanism of Bioactive Compounds from Actinobacteria against Drug-Resistant Pathogens

Once scientists provide new antimicrobial drugs, the microorganism starts to adapt themselves against these drugs, which become ineffective at some points. This is primarily due to changes that occur inside the microorganism, especially bacteria, due to the interaction of numerous organisms through their environment and surroundings. These changes may occur for various reasons: mutations, selective pressure, gene transfer and phenotypic change [65]. For example, gene mutation occurs when bacteria reproduce, leading to the development of bacteria with genes that help them resist antibiotics. In addition, the mutation leads to alter the target and modification of the drug-receptor site in the target site [66].

Moreover, selective pressure means that bacteria carrying resistance genes hold up and multiply so that new resistance bacteria become the predominant type. The selective pressure leads to a lack of entry and decreased cell permeability. In addition, it leads to greater exit and active efflux pump [67]. Bacteria unnecessarily replicate to transmit their antibiotic resistance gene. Instead, it is passed across various types of bacteria resistance determinants through horizontal gene transfer making the bacterium resistant [68]. Besides this, phenotypic change suggests that the bacteria can change some of their properties to become more resistant to common antibiotics using the enzymatic inactivation of the antibiotics or by synthesising resistant metabolic pathways [69]. Other mechanisms, by modulation of methicillin-resistance of penicillin-binding protein (PBP2a) synthesis, were regulated by two genes known as MecI and MecR1 proteins. When existing, the signalling or regulatory proteins of the plasmid-mediated staphylococcal β-lactamase gene bla-Z system are working.

Furthermore, homogeneous tolerance is based on mutations at a different locus of genetics. Furthermore, other external and internal causes affect the development of methicillin resistance [115]. The mechanism of anti-bacterial resistance is demonstrated in Figure 6.

## 9. Conclusions and Future Prospects

There is a global demand for new chemotherapeutic agents and antibiotics that are extremely active and have low toxicity and environmental effects. The drug resistance in viruses, fungal and bacteria, as well as the emergence of life-threatening microorganisms, have become higher than before. This is due to the wrong usage of dose and time of medication administration, increasing the requirement of new and active compounds that assist and relieve all the aspects of the human condition. Actinobacteria have been isolated from different ecosystems, including several medicinal plants from the terrestrial and rhizosphere environment, hot springs as thermophilic Actinobacteria, deep-sea sediments, marine sponges and alkalines line soil. Several previous published works reported that Actinobacteria are understudied phylogenetic groups with high biosynthetic potential. This type of bacteria was found to have the greatest number of biosynthetic gene clusters in its genomes. There are more opportunities to investigate new resources and other biological characteristics of previously inaccessible natural products extracted from Actinobacteria as interest grows in bioactive molecules from drug formulation with minimal side effects. This group represents the new resource of bioactive compounds due to its ability to grow in multisectoral environments.

This review focused on the emphasis and the connection between finding new resources and novel strategies to search for potential bioactive compounds isolated from phylum Actinobacteria. In addition, this review highlighted some limitations in today’s research regarding a new source for bioactive compounds isolated from Actinobacteria. For example, most previous publications concentrated only on the endophytic Actinobacteria, excluding other environments such as thermophilic, alkaliphilic and haloalkaliphilic Actinobacteria. Note that many other biodiversities and multidisciplinary environments represent important and new resources of potentially bioactive compounds. As a result, it appears that certain new techniques and methodologies for a thorough investigation of bioactive natural compounds are required. This includes, for example, the discovery of novel structure–activity relationships in nature, which has become increasingly important for the synthesis inspiration of natural bioactive product compounds. This results in increased diversity with less complexity and a good knowledge of isolation processes.

To address the challenges of biodiversity and promote future sustainable use of natural resources, a multidisciplinary perspective is required to find, describe and convey nature’s richness. This is necessary for identifying novel bioactive chemicals from Actinobacteria. Moreover, this review summarised the new bioactive compounds isolated from Actinobacteria and their applications in industrial, agricultural and environmental protection, pharmaceutical bioactive compounds and pharmaceutically related biomolecules. This includes superordinate metabolites that act as inhibitory or killing agents against pathogens that affect humans and animals, including resistant bacteria, fungi, viruses and several protozoa. They can also produce an anti-cancer and several enzymes for active degradation and meeting industrial demands worldwide. Therefore, continuous selective isolation and screening studies on the characterisation and identification of novel potential bioactive compounds from Actinobacteria are required. This is to create commercially viable, long-term and cost-effective production methods. Their metabolic flexibility and abundance also provide a novel, strong pathway for the bioremediation of organic wastes and contaminants.

## Figures and Tables

**Figure 1 molecules-26-04504-f001:**
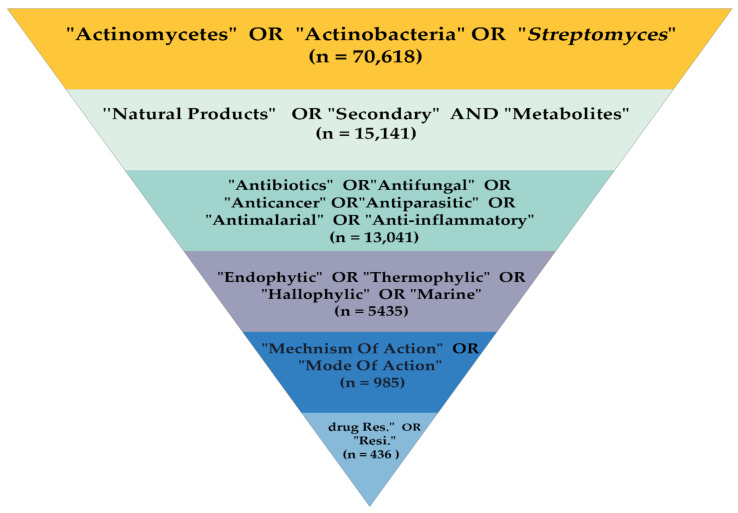
Phases of the review protocol. The process involved six main steps using the general keywords “Actinomycetes” OR “Actinobacteria,” OR “*Streptomyces”*.

**Figure 2 molecules-26-04504-f002:**
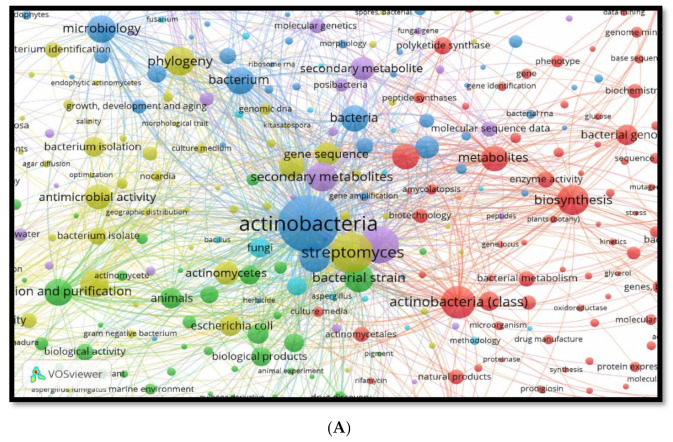
VOSviewer software tool to analyse and visualise scientific literature from 2010 to 2021 for phylum Actinobacteria. (**A**) Diversity of keywords from the Scopus database for *Streptomyces*, Actinobacteria, natural products, primary, secondary metabolites, habitat effects of environments, pharmaceutical industry; (**B**) Spread of reviewed and cited papers based on the dispersed keywords of anti-infective agents and occurrence. (**C**) The countries with the highest numbers of publication and citation. N.B: The name of countries in small letters from the VOSviewer software itself.

**Figure 3 molecules-26-04504-f003:**
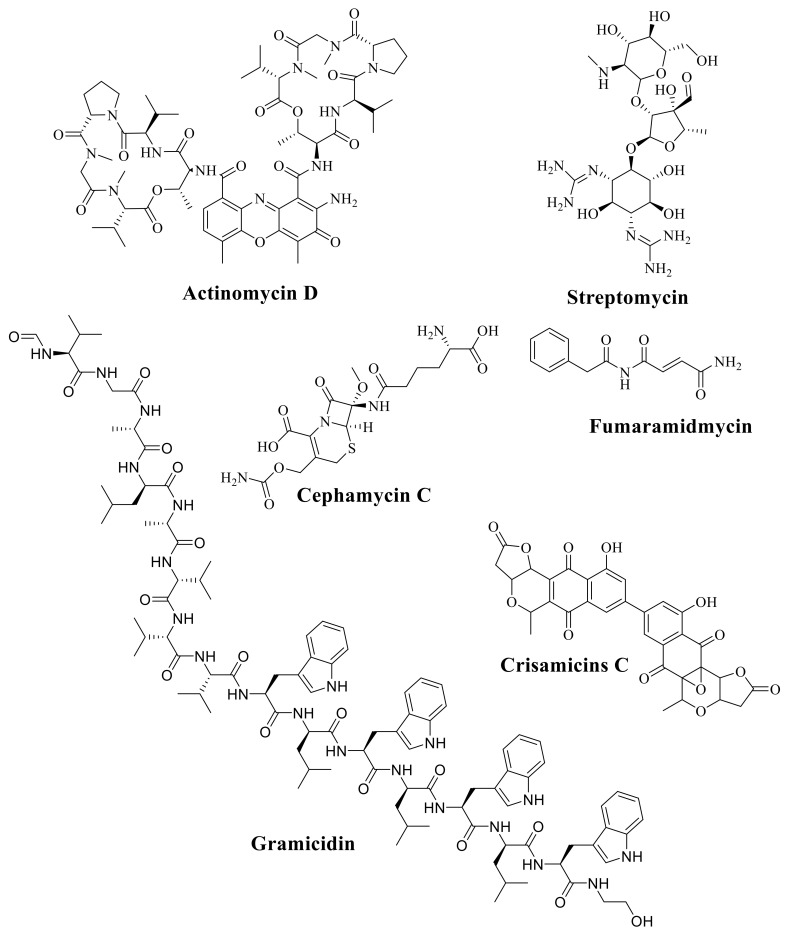
The molecular structure of Actinomycin D; Streptomycin; Gramicidin; Cephamycin C; Fumaramidmycin; and Crisamicins C.

**Figure 4 molecules-26-04504-f004:**
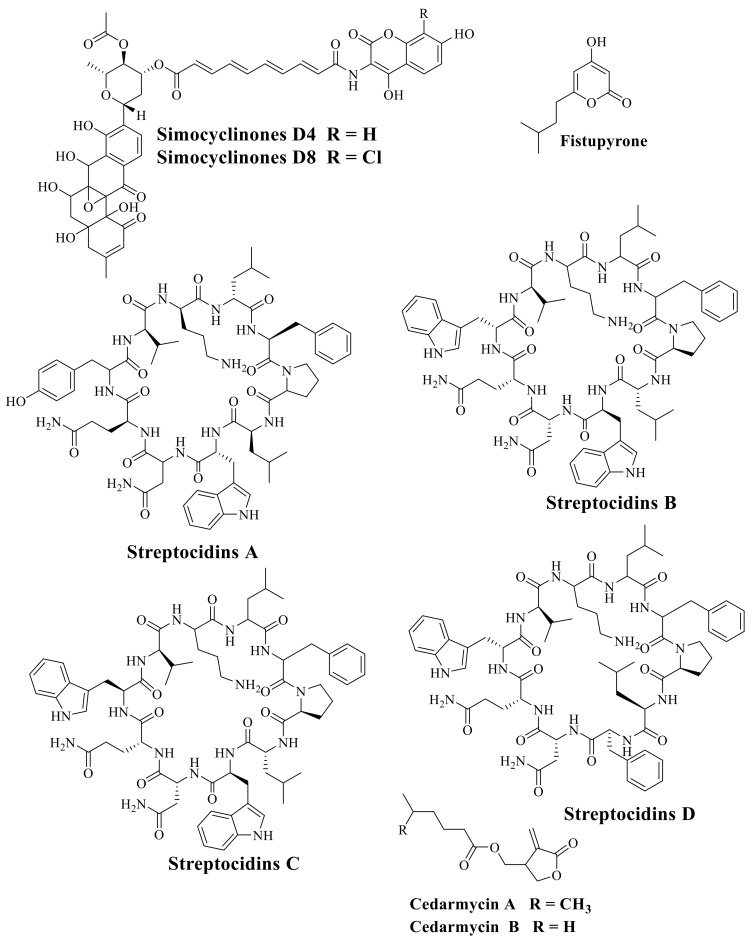
The molecular structure of Simocyclinones D4 and D8; Fistupyrone; Streptocidins A–D; Cedarmycin A and B.

**Figure 5 molecules-26-04504-f005:**
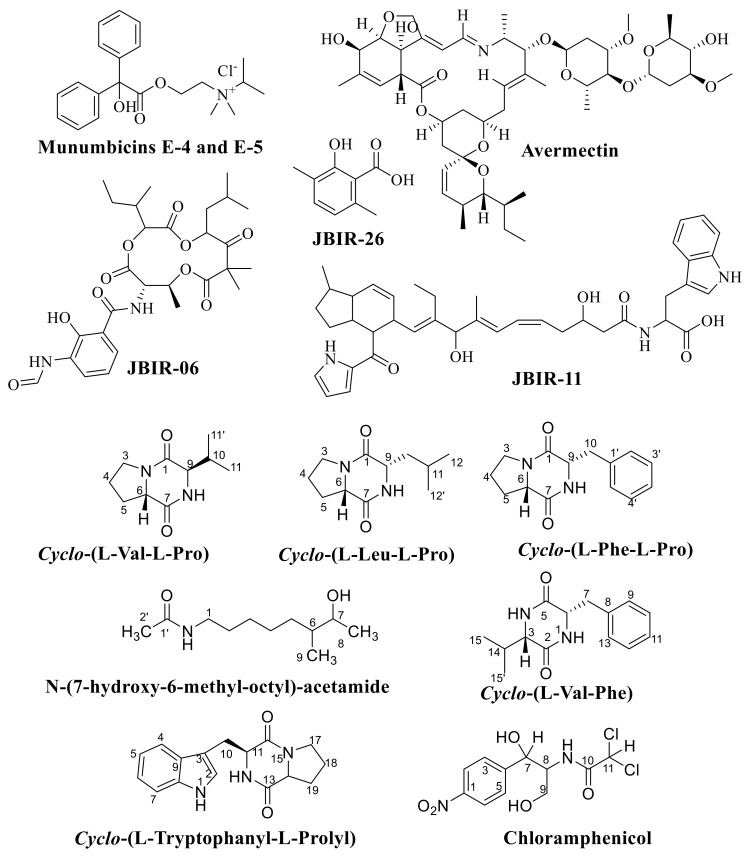
The molecular structure of Munumbicins E-4 and E-5; Avermectin; JBIR-06; JBIR-11; and JBIR-26. *Cyclo*-(L-Val-L-Pro), *Cyclo*-(L-Leu-L-Pro), *Cyclo*-(L-Phe-L-Pro), N-(7-Hydroxy-6-Methyl-Octyl)-Acetamide, *Cyclo*-(L-Val-L-Phe), *Cyclo*-(L-Tryptophanyl-L-Prolyl) and Chloramphenicol.

**Figure 6 molecules-26-04504-f006:**
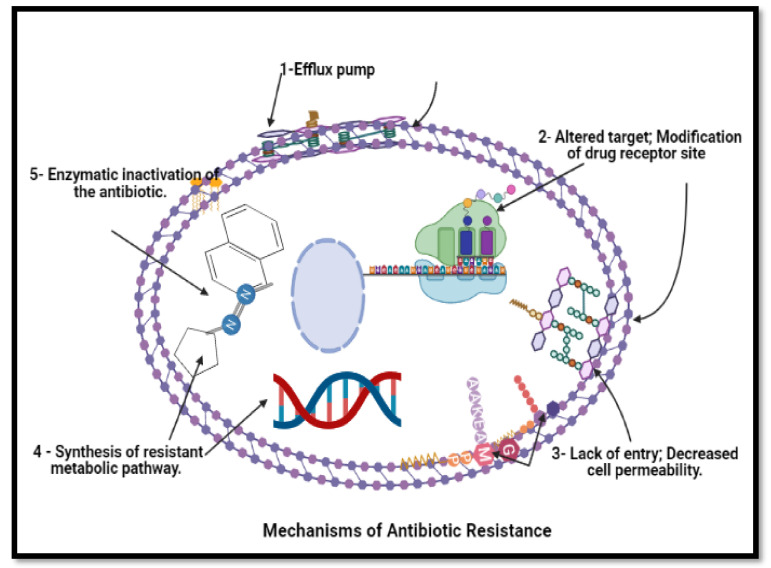
Mechanism of anti-bacterial resistance to avoid killing by antimicrobial molecules. N.B.: BioRender was used to draw these scientific figures.

**Table 1 molecules-26-04504-t001:** List of bioactive compounds, producers, source of isolation and screening for their bioactivity isolated from aquatic and marine Actinobacteria.

Bioactive Compound	Producer	Chemical Group	Bioactivity	Source of Isolation	References
Unknown	M72, M1, M71, W38, W108 and M93	N/A	Anti-bacterial	Chenab River Sediments	[29]
Lynamicins, spiroindimicins	*Streptomyces sp.*	Bisindole pyrrole	Anti-bacterial	Deep sea marine sediment	[32]
Anandins	*Streptomyces anandii*	Steroidal Alkaloids	Cytotoxic	Marine sediments from mangrove zone	[33]
Paulomycin G	*Micromonospora matsumotoense*	Paulomycin derivatives	Anti-tumor properties	Deep sea marine sediment	[34]
Rifamycin B	*Salinispora sp.*	Polyketides	Anti-bacterial	Sediment	[35]
Manzamine A	*Micromonospora sp.*	Alkaloid	Antimalarial	Symbiont to sponge Acanthostrongylophora	[36]
Violapyrone B	*Streptomyces somaliensis*	*α-pyrone*	Anti-bacterial	Deep sea marine sediment	[37]

Abbreviation: S. = (*Streptomyces),* N/A; not applicable.

**Table 2 molecules-26-04504-t002:** List of bioactive compounds from thermophilic, alkaliphilic and haloalkaliphilic Actinobacteria.

Bioactive Compound	Producer	Chemical Class	References
Thermomycin	*Streptomyces* *thermophilus*	Polyketide Antibiotic	[40]
Anthramycin	*Streptomyces* *refuineus*	Benzodiazepine Alkaloid	[40]
Pyridine-2,5-diacetamide	*Streptomyces* sp. DA3-7	Antimicrobial	[48]
1, 4-butanediol, adipic acid, & terephthalic acid	*Thermomonospora fusca*	aliphatic-aromatic copolyesters	[49]

**Table 4 molecules-26-04504-t004:** List of anti-MRSA activity of bioactive compounds, producers and their chemical classes isolated from Actinobacteria.

Antibiotic	Producer	Chemical Class	Reference
Angumicynones A (1); Angumicynones B (2); Angucyclinones analogues compounds 3–8	*Streptomyces* sp. MC004	Angucyclic quinones	[94]
Watasemycin A (3)		Thiazostatins	
Pulicatin G (4) and aerugine (5)		Benzyl thiazole and thiazoline	
Polyketide [2-hydroxy-5-((6-hydroxy-4-oxo-4H-pyran-2-yl) methyl)-2-propylchroman-4-one]	*Streptomyces* *sundarbansensis* WR1L1S8	N/A	[95]
Azalomycin F5a (1) and its four derivative compounds:	*Streptomyces hygroscopicus* var. *azalomyceticus*	Polyhydroxy macrolide	[96]
Gargantulide A	*Streptomyces* sp. A42983	Macrolactone	[97]
New Ikarugamycins: Compound 1: Isoikarugamycin; Compound 2: 28-N-methylikarugamycin; Compound 3: 30-oxo-28-N-methyl-ikarugamycin; Compound 4: Ikarugamycin; Compound 5: MKN-003B; Compound 6: 1 H-indole-3-carboxaldehyde; Compound 7: Phenylethanoic acid	*Streptomyces zhaozhouensis* CA-185989	Compounds 1- 4: Pentacyclic tetramic acid macrolactams; Compound 5: Butenolide; Compound 6: Indole; Compound 7: Acetic acid	[98]
Pyrrole-Like Structure	*Streptomyces* sp. MN41	pyrrole	[99]
actinomycins V, X2 and D.	*Streptomyces* antibioticus NBRC 12838^T^	Actinomycins	[100]
Abyssomicin C	Actinobacteria	polyketide	[93]
laidlomycin	*Streptomyces sp. CS684*	affecting the metabolism	[89]
Neocitreamicins I and II	*Nocardia*		[93]
Etamycin	Actinomycetes strains CNS-575	cyclic peptide	[101]
Dichloromethane	Actinobacteria (I-400A, B1-T61, M10-77)	N/A	[93]
2, 4-dichloro-5-sulfamoyl benzoic acid (DSBA)	*Streptomyces* sp. VITBRK2	N/A	[93]

Abbreviation: *S.* = (*Streptomyces),* N/A; not applicable.

**Table 5 molecules-26-04504-t005:** List of enzymes, their producers, uses and application in industry isolated from Actinobacteria.

Enzyme	Producer	Use	Application in Industry	References
**Protease**	*Thermoactinomyces sp.,*	Detergents	Detergent	[106]
	*Nocardiopsis* *sp.,* *S*	Cheese making	Food	[106]
	*Pactum, Streptomyces*	Clarification—low-calorie beer	Brewing	[107]
	*Hermoviolaceus, S**Sp*.	Dehairing	Leather	[107]
**Cellulase**	*S. Thermobifida*	Removal of stains	Detergent	[108]
	*Halotolerans, S. Sp., Ruber*	Denim finishing, softening of cotton	Textile	[106]
		Deinking, modification of fibres	Paper and pulp	[108]
**Lipase**	*S. griseus*	Removal of stains	Detergent	[109]
		Stability of dough and conditioning	Baking	[109]
		Cheese flavouring	Dairy	[110]
**Xylanase**	*Actinomadura Sp.*	Conditioning of dough	Baking	[110]
		Digestibility	Animal feed	[111]
		Bleach boosting	Paper and pulp	[111]
**Pectinase**	*S. lydicus*	Clarification, mashing	Beverage	[112]
		Scouring	Textile	[113]
**Amylase**	*S. erumpens*	Deinking, drainage	Paper and pulp	[114]
		Removal of stains	Detergent	[114]

Abbreviation: *S.* = (*Streptomyces).*

## Data Availability

Appendix A related to this article can be found in the Supplementary data file.

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
