# Peer review of "Biodiversity of Secondary Metabolites Compounds Isolated from Phylum Actinobacteria and Its Therapeutic Applications"

_molecules, 2021, doi:10.3390/molecules26154504_

Round 1

Reviewer 1 Report

In this manuscript  the authors tried to demonstrate the biodiversity of secondary metabolites compounds isolated from phylum actinobacteria and its potential applications. The following are my comments and critique:

# The manuscript is easy to read and understand. However, there are some grammatical and punctuation errors and in some instances, the manuscript need to improve. (for example: Compounds “abstract part (Line, 24)” C was capital; replace extracted by “gained” in abstract part (Line, 31); It is vs its please unify within the whole manuscript, the authors need to make space between end of paragraph and new words/values. and so on.. check all manuscript.

# Actinobacteria is a group of diverse bacteria, and most species in this class of bacteria are filamentous aerobes found in soil, including the genus Streptomyces. (Line, 35) Why the authors said “Actinobacteria and Streptomyces have a large number…….” in abstract part ? please correct it.

# Please, normalize the writing style of the heading and subheading e.g. “2.1. Study design and Search strategy”, letter S in search should be small or capital (Line 94). Please, do it in all manuscript. I found many different styles

# Please define abbreviations upon first appearance in the text (specially names of microbes), correct it.

#In the manuscript, the authors mentioned many of abbreviations, please make a list of abbreviations and insert it in the appropriate position according to journal style.

Author Response

Thank you for giving us the opportunity to submit a revised manuscript entitled

“Biodiversity of Secondary Metabolites Compounds Isolated from Phylum Actinobacteria and its Therapeutic Applications” for publication in the Journal Molecules. We appreciate the effort that you and the reviewer’s feedback and are grateful for the insightful comments to improve the manuscript. We have incorporated and responded to all suggestions made by the reviewers. Those changes are highlighted within the manuscript. Please see below, in red, for a point-by-point response to the reviewers’ comments and concerns. All page numbers refer to the revised manuscript file with tracked changes.

Author's Reply to the Review Report (Reviewer 1)

Comments and Suggestions for Authors

In this manuscript the authors tried to demonstrate the biodiversity of secondary metabolites compounds isolated from phylum actinobacteria and its potential applications. The following are my comments and critique:

  • The manuscript is easy to read and understand. However, there are some grammatical and punctuation errors and in some instances, the manuscript need to improve. (for example: Compounds “abstract part (Line, 24)” C was capital; replace extracted by “gained” in abstract part (Line, 31);
  • Author response: Corrected now as recommended by reviewer.
  • Compounds corrected and added gained.

  • It is vs its please unify within the whole manuscript, the authors need to make space between end of paragraph and new words/values. and so on. check all manuscript.
  • Author response: All it is and it’s were corrected. Its in this MS is a possessive pronoun meaning, "belonging to it,
  • Actinobacteria is a group of diverse bacteria, and most species in this class of bacteria are filamentous aerobes found in soil, including the genus Streptomyces. (Line, 35) Why the authors said “Actinobacteria and Streptomyces have a large number…….” in abstract part? please correct it.
  • Author response: and Streptomyces is deleted
  • Please, normalize the writing style of the heading and subheading e.g. “2.1. Study design and Search strategy”, letter S in search should be small or capital (Line 94). Please, do it in all manuscript. I found many different styles.
  • Author response: Corrected all sections and subsection corrected as journal style.
  • Line 101: Study Design and Search Strategy.
  • Line 175: Actinobacteria
  • LINE 251: Thermophilic Actinobacteria
  • Please define abbreviations upon first appearance in the text (specially names of microbes), correct it.
  • Author response: The abbreviations of microbes are added under the tables and the name of microbes write as first time in complete name.
  • V= Vibrio vulnificus, Vibrio parahemolyticus and Vibrio harveyi. Line 225
  • (MIC)= Minimum inhibitory concentration LINE 231.
  • (MRSA): Methicillin-resistant Staphylococcus aureus
  • (VRE)Vancomycin-resistant Enterococcus
  • (ATCC): American Type Culture Collection
  • (PBP2a): penicillin-binding protein
  • S. Streptomyces
  • Some very common abbreviations do not need to be defined.
  •  
  • In the manuscript, the authors mentioned many of abbreviations, please make a list of abbreviations and insert it in the appropriate position according to journal style.
  • Author response: The abbreviations of microbes are added under the tables and the name of microbes write as first time in complete name. Abbreviations were added under the tables other abbreviations were written in the MS for defined the first time.

Reviewer 2 Report

Please find attached pdf.

Author Response

Thank you for giving us the opportunity to submit a revised manuscript entitled

“Biodiversity of Secondary Metabolites Compounds Isolated from Phylum Actinobacteria and its Therapeutic Applications” for publication in the Journal Molecules. We appreciate the effort that you and the reviewer’s feedback and are grateful for the insightful comments to improve the manuscript. We have incorporated and responded to all suggestions made by the reviewers. Those changes are highlighted within the manuscript. Please see below, in red, for a point-by-point response to the reviewers’ comments and concerns. All page numbers refer to the revised manuscript file with tracked changes.

Author's Reply to the Review Report (Reviewer 2)

The presented manuscript named „Biodiversity of Secondary Metabolites Compounds Isolated from Phylum Actinobacteria and its Therapeutic Applications: A review„ is a comprehensive study with potential applications and should not be neglected regarding future research in the area of compounds isolated from Actinobacteria. Although the presented manuscript could be considered for the special issue „Natural Products as Antimicrobial Agents: From Extraction to Therapeutic Applications However”, generally, in my opinion, the novelty and scientific significance of the presented work is weak for both special issue and the journal Molecules MDPI.

Some issues must be addressed to improve work presentation.

Abstract:

L 23-25

  • The current review aims to summarise the biosynthesis of novel secondary metabolites Compounds, biodiversity of the phylum actinobacteria and the diverse range of secondary metabolites produced that vary depending on the ecological environments they inhabit. „ Please reformulate, as it stated it seems that secondary metabolites inhabit different ecological environments.
  • Author response: The current review aims to summarise the biodiversity and biosynthesis of novel secondary metabolites compounds, of the phylum actinobacteria and the diverse range of secondary metabolites produced that vary depending on its ecological environments they inhabit.

Introduction:

  • I would suggest authors improve and reorganize this part of the manuscript.

L 58 Please provide a reference.

  • Author response: The reference is added.

  • L 60 Is it possible to explain better the classification of Streptomyces?
  • Author response: Line: 71-77:

(The genus Streptomyces is a Gram-positive bacterium, it is the largest genus of the phylum actinobacteria, which have a complex growth and can produce various secondary metabolites [9]. In addition, at least there are more than 800 Streptomyces species may be found until date (see http://www.bacterio.net/Streptomyces.html) [10]. Streptomyces is the major microbial genus of the most antibiotic-producing bacteria in the microbial world discovered so far, where streptomycin, gentamycin, rifamycin, chloramphenicol, and erythromycin are produced by Streptomyces [11]).

  • L 67-68

“They are found in many habitats, ranging from endophytically colonising host plants’

intracellular space, such as terrestrial and aquatic environments.” Please reformulate.

Author response: Line 67-70:

Actinobacteria have ubiquitous characteristics. They are present in diverse ecosystems on the earth such as endophytically with plants and in terrestrial and aquatic environments.

  • L 85-86

“This also includes the effects of environments and ecosystem on the phylum actinobacteria

producing potential bioactive compounds in the pharmaceutical industry market.” Please

reformulate, there can be no effects from the environments and ecosystem on phylum.

  • Author response: In addition, to determine the effects of environments and ecosystem on the phylum actinobacteria that producing potential bioactive compounds in the pharmaceutical industry market.

  • L 89-90

“…and isolate specific bioactive compounds from different ecosystems” Please be more

specific. It is not possible to isolate active compounds from ecosystems.

  • Author response: Corrected as (This information would be helpful to other scholars who attempt to discover and isolate specific bioactive compounds from phylum actinobacteria under different ecosystems)

  • In general, I would suggest authors consider the reorganization of the Introduction. In my

opinion, the logical order of presented backup information’s would be 1) Line 45-56 2) 67-82

3) 57-66 4) 83-92 Lines: 46- 99

-  Author response: The introduction was corrected and reorganized as the reviewer suggest.

  1. Methods and Protocols
  • Please explain used VOSviewer.
  • Author response: The VOSviewer software was used to visualise the bibliometric networks to build assistance for common bibliographic databases from the Scopus database. The most important journals, articles, authors, organisations and states among such articles were identified. Besides, bibliographic coupling and most utilised keywords in the abstracts with keywords and titles were identified.

  • L 132-136 this should be moved to chapter 3.
  • Author response: (Actinobacteria synthesise diverse metabolite molecules that have key roles in their heterogeneous and complex microenvironments. Natural products, also known as secondary metabolites, are useful compounds developed by microbes. These are not usually needed for natural cell development, but they provide advantages to the cells in other ways [13]. Such compounds could play roles in inhibition, communication, nutrient acquisition, or other associations with nearby environments or organisms). move to chapter 3 lines 157- 162.

  • Microbial Ecology and Actinobacteria

L 208 Please replace “produce” with the appropriate term. The anti-tumor properties cannot be produced.

  • Author response: it was exhibited bioactive compounds with anti-tumour properties

  • L 209 “antimalarial” is activity or effect, therefore cannot be “reported as bioactive

compounds”. Please correct.

  • Author response: Line 222 -223: antimalarial activity was also reported as bioactive compounds from actinobacteria as bioactive compounds by Fukuda et al. [35].

  • L 223 “Some bioactive compounds isolated from aquatic and marine environments…” once again, compounds cannot be isolated from environments.
  • Author response: List of bioactive compounds isolated from aquatic and marine actinobacteria aquatic and marine environments are demonstrated in (Table 1).

  • L 225 Table 1. “…and screening for their antimicrobial activity” Please correct since in table1 anti-tumor and cytotoxic activities are also presented.
  • Author response: Table 1. list of bioactive compounds, producers, source of isolation and screening for their bioactivity isolated from aquatic and marine actinobacteria.

  • The important antibiotics isolated from actinobacteria
  • Author response: Table 3. The list of important producers and chemical classification of antibiotics isolated from actinobacteria.
  •  
  • L 305 Candida albicans and Mucor miehei – Italic
  • Author response: Line 320: Candida albicans and Mucor miehei, corrected
  •  
  • L 316 Please insert “list” between antibiotic and separated.
  • Author response: The updated list of antibiotics separated from actinobacteria and Streptomyces is given in Table 3.
  •  
  • L 334 Table 3. “... and the application isolated from” Please reformulate.
  • Author response: Line 350; Table 3. The list of important producers and chemical classification of antibiotics isolated from actinobacteria.

Reviewer 3 Report

The review is interesting but it needs several corrections:

Title: please remove “a review”.

Line 35, 60 an and following: being Streptomyces the largest genus of Actinobacteria it is inappropriate to write “Actinobacteria and Streptomyces”.

Keywords: it is useless to repeat words already included in the title

Line 165, 174 and following: Actinobacteria must have the first A upper case.

Line 97 and following: for 2021 scientific literature, it is necessary to indicate up to which month of the year (or exact date) the Scopus search carried out.

Line 332: please add the reference.

Table 4: it is opportune to separate with a line (or a pale background) the data related to every single reference, otherwise the table is difficult to read.

Line 360: enzymes are not secondary metabolite for bacteria, please delete “secondary metabolite”.

Lines 362-363: the enzyme name are normally written lower case.

Figure 6A should be omitted because the interesting elements are included in Figure 6B.

In the References section please write all species names in Italics.

Author Response

Thank you for giving us the opportunity to submit a revised manuscript entitled

“Biodiversity of Secondary Metabolites Compounds Isolated from Phylum Actinobacteria and its Therapeutic Applications” for publication in the Journal Molecules. We appreciate the effort that you and the reviewer’s feedback and are grateful for the insightful comments to improve the manuscript. We have incorporated and responded to all suggestions made by the reviewers. Those changes are highlighted within the manuscript. Please see below, in red, for a point-by-point response to the reviewers’ comments and concerns. All page numbers refer to the revised manuscript file with tracked changes.

Author's Reply to the Review Report (Reviewer 3)

Comments and Suggestions for Authors

  • The review is interesting but it needs several corrections:
  • Author response: Thank you so much.
  • Title: please remove “a review”.
  • Author response: Deleted from the title.
  • Line 35, 60 an and following: being Streptomyces the largest genus of Actinobacteria it is inappropriate to write “Actinobacteria and Streptomyces”.
  • Author response: Deleted and Streptomyces
  • Keywords: it is useless to repeat words already included in the title
  • Author response: Keywords: Microbial Ecology; Aquatic and Marine Environments; Drug-Resistant Pathogens; Streptomyces; Natural Products; VOSviewer software.
  • Line 165, 174 and following: Actinobacteria must have the first A upper case.

Author response: Corrected in line 176.

  • Line 97 and following: for 2021 scientific literature, it is necessary to indicate up to which month of the year (or exact date) the Scopus search carried out.
  • The date is added (2010 to March, 22nd ,2021) line 29 and 104.
  • Line 332: please add the reference.
  • Author response: Furthermore, Streptomyces sp. PVRK-1 was isolated from the Manakkudy mangroves in Tamil Nadu, South India. This Streptomyces has been able to produce many small molecules exhibiting potent an-ti-MRSA activity [59].

  • Table 4: it is opportune to separate with a line (or a pale background) the data related to every single reference, otherwise the table is difficult to read.
  • Author response: Table 4 and Table 5 separate with a line as you recommended.
  • Line 360: enzymes are not secondary metabolite for bacteria, please delete “secondary metabolite”.
  • Author response: Yes. You are correct. thanks for this comment. It is change to (Production of Enzymes from Actinobacteria)

  • Lines 362-363: the enzyme name is normally written lower case.
  • Author response: Line 382-383: They can secrete protease, cellulases, lipase, xylanase, pectinase, and amylase.
  •  
  • Figure 6A should be omitted because the interesting elements are included in Figure 6B.
  • Author response: Figure 6A omitted. Now it is only Figure 6:

Figure 6. Mechanism of anti-bacterial resistance to avoid killing by antimicrobial molecules.

  • In the References section please write all species names in Italics.
  • Author response: Thank you for this comments: The references are corrected as we detected some species names are not in Italics.

  • Extra corrections:

  • LINE 88-91: VOSviewer as it may be used to create a networks of scientific journals, researchers, key-words or terms, countries, scientific publications, research organizations. Items in these networks can be linked by citation, bibliographic coupling, co-authorship, co-citation or co- occurrence links.
  • Table 4. List of anti-MRSA activity of bioactive compounds, producers and its chemical classes isolated from

Round 2

Reviewer 2 Report

The manuscript is significantly improved. Technical checking and improvement of English are mandatory before publication.

Reviewer 3 Report

Just a few formal comments:
Actinobacteria is a phylum so it should be written in standard font with the first A capitalized;
in the first part of the text too often the word Actinobacteria is associated with Streptomyces, forgetting that Streptomyces is a part of Actinobacteria;
remove the italics from Actinobacteria in Table 4 (ref. 97) and Streptomycetes which is not a genus.